# Characterization of Grain Size in 316L Stainless Steel Using the Attenuation of Rayleigh Wave Measured by Air-Coupled Transducer

**DOI:** 10.3390/ma14081901

**Published:** 2021-04-11

**Authors:** Meng Wang, Yangguang Bu, Zhaojie Dai, Shengyang Zeng

**Affiliations:** School of Traffic and Transportation Engineering, Central South University, Changsha 410075, China; MengWang1995@csu.edu.cn (M.W.); buyangguang@csu.edu.cn (Y.B.); csu_dzj@163.com (Z.D.)

**Keywords:** grain size, 316L stainless steel, Rayleigh wave attenuation, air-coupled transducer, heat treatment

## Abstract

Grain size is an important parameter in evaluating the properties of microstructures in metals. In this paper, the attenuation coefficient of Rayleigh waves is introduced to characterize grain size in heat treated 316L stainless steel. Rayleigh wave attenuation is measured using an angle beam wedge transducer as the transmitter and an air-coupled transducer as the receiver. The results show that the grain size in 316L stainless steel increases due to heat treatment time, the hardness decreases accordingly, and the attenuation coefficient of Rayleigh waves increases. This indicates that the Rayleigh wave attenuation is sufficient in distinguishing the changes in the properties of the heat-treated stainless steel. It is found that compared with the measurement method using an angle beam wedge receiver, the measured results are efficient, more stable and less influenced by the surface state when an air-coupled receiver is used. In addition, comparison results also show that the Rayleigh wave attenuation is more sensitive to changes in material properties than the longitudinal wave attenuation, as the wavelength of the Rayleigh wave is shorter than that of the longitudinal wave at the same frequency.

## 1. Introduction

Grain size is an important parameter in characterizing the microstructure of metals and has a great effect on properties of metal, such as yield strength, plasticity, toughness, fatigue strength, creep strength, and corrosion resistance [1]. However, processes, such as melting, heat treatment, or machining can result in grain sizes that deviate from the design specifications [2]. Therefore, accurate characteristic of the grain size is essential to monitoring the mechanical properties.

Traditionally, a small piece of specimen is taken from material and polished for microstructure imaging with optical and electron microscopy [3]. Though this method provides extensive and accurate details, it is constrained by their destructive nature, as well as laboratory confinement and time-consuming procedures for specimen preparation [4]. The ultrasonic nondestructive methods are widely used due to its high sensitivity to changes in material internal condition. The scattered wave from material microstructure contains the grain size information, which are indicated in ultrasound dispersion [5], attenuation [1,6,7,8] and backscattered grain noise [9,10,11,12].

The backscatter grain noise has no requirement of parallel front-wall and back-wall surfaces and known wall thickness of a specimen, but more complicated statistical methods are required to combine the ultrasonic signals from different sets of grains at different transducer spatial positions [13]. This method loses accuracy in the case of coarse-grained materials. Meanwhile, the immersed ultrasonic technique is generally used with focus transducers to enhance the backscatter signals, which is limited by the large size, as well as immovability and service status of the workpiece in industrial applications. At present, the scattering model is established to associate the grain size with the longitudinal wave attenuation, which makes one may achieve quantitative results of grain size with precise attenuation (longitudinal or shear) measurements [14,15,16]. However, the longitudinal wave attenuation in a low frequency range is too small in fine-grain materials to recognize the variation of the grain size, and multiple reflections, well surface condition and necessary thickness are needed to accurately determination [17]. The shear wave attenuation measurement process is more complicated and the measurement results are easily affected by the coupling state, as well as the contact pressure [6,18]. Non-linear methods are gradually emerging in grain size evaluation due to their high sensitivity to changes of material microstructures, but they have the same shortcomings as shear wave attenuation since only contact transducers can be used. In addition, the signal is susceptible to the influence of precipitated phases, residual stress in material [4]. A more flexible and practical technique is hence required for industrial application.

The Rayleigh wave can propagate a long distance with comparable attenuation to longitudinal wave at the similar frequency and is not affected by the thickness of the specimen because its energy is mainly concentrated on the surface of the material [19,20]. In addition, the measurements can be implemented with access from one side efficiently with an air-coupled receiver, which can significantly reduce the negative effects of surface conditions. Despite of above apparent advantages in practice compared to other methods, few literatures exist that review characterizing grain size with Raleigh wave, since only the grains at the material surface are involved in the measurement. Although surface-layer grains may not represent the characteristic of grain distributions of the whole sample, they have strong correlations with the changes of material microstructure induced by external conditions such as high temperature, high pressure and impact load, as well as surface treatment techniques, such as surface shot peening and laser burnish. Therefore, Rayleigh wave attenuation can be an option in characterizing the properties of the component in practice and it is worth to investigate its feasibility of grain size characterization.

In this work, the 316L stainless steel specimens with different heat treatment processes are prepared for different grain size. The Rayleigh waves experiments are conducted with both an air-coupled receiver and a contact receiver to investigate the effects of surface condition, and the attenuation coefficient is extracted using proposed method by fitting the experimental results to theoretical predictions with consideration of beam diffraction. The relationship between the grain size and the Rayleigh wave attenuation coefficient are discussed.

## 2. Measurement Method and Theory

A schematic diagram for measuring the attenuation coefficient using a contact wedge transmitter and an air-coupled receiver is shown in Figure 1. The longitudinal waves in the wedge generated by the transducer propagate into the specimen at an angle $\theta_{1}$. Surface waves are subsequently generated and propagate along the surface of the specimen, transmit to the air and will be finally received by the air-coupled transducer at an angle $\theta_{2}$. According to Snell’s law, $\theta_{1}=\arcsin\left(c_{r}/c_{p1}\right)$ and $\theta_{2}=\arcsin\left(c_{p2}/c_{r}\right)$, where $c_{r}$ is the Rayleigh wave velocity in the specimen, $c_{p1}$ is the longitudinal wave velocity in the wedge, and $c_{p2}$ is the longitudinal wave velocity in air. The coordinate system $x_{1}y_{1}z_{1}$ is located on the surface of the angle beam wedge transducer, $x_{2}y_{2}z_{2}$ is on the contact surface of the wedge and the specimen, and $x_{3}y_{3}z_{3}$ is on the surface of the air-coupled transducer.

Considering the generation of ultrasonic Rayleigh waves on the contact surface of the wedge and the specimen, the propagation on the surface of the specimen and the leakage into the air, and being received by the air-coupled transducer, the Rayleigh wave sound pressure distribution at different distances can be obtained by the Sommerfeld-Rayleigh integral method [21],
(1)$$u\left(x\right)=u_{0}\int_{S_{r}}\int_{S^{\prime }}\int_{S_{F}}\int_{S_{T}}\frac{\exp\left(ik_{p1}R_{1}\right)}{R_{1}}\frac{\exp\left(ik_{r}R_{2}-\alpha_{r}R_{2}\right)}{\sqrt{R_{2}}}\frac{\exp\left(ik_{p2}R_{3}\right)}{R_{3}}dS_{r}dS^{\prime }dS_{F}dS_{T}$$
where $u_{0}$ is the initial displacement amplitude, $R_{1}=\sqrt{{\left(x_{1}-x_{1}^{\prime }\right)}^{2}+{\left(y_{1}-y_{1}^{\prime }\right)}^{2}+{\left(z_{1}\right)}^{2}}$ indicates the distance from the angle beam wedge transducer to the elliptical sound source under the wedge, $S_{T}$ is the area of the angle beam wedge transducer, $k_{p1}=\omega /c_{p1}$ is the wavenumber in the wedge, and ω is the angular frequency. $R_{2}=\sqrt{{\left(x_{2}-x_{2}^{\prime }\right)}^{2}+{\left(y_{2}-y_{2}^{\prime }\right)}^{2}}$ indicates the distance the sound source travels along the surface of the specimen, $S_{F}$ is the area of the semi-elliptical sound source, $k_{r}=\omega /c_{r}$ is the wavenumber on the surface of the specimen, and $\alpha_{r}$ is the ultrasonic Rayleigh wave attenuation coefficient. $R_{3}=\sqrt{{\left(x_{3}-x_{3}^{\prime }\right)}^{2}+{\left(y_{3}-y_{3}^{\prime }\right)}^{2}+{\left(z_{3}\right)}^{2}}$ indicates the distance from the surface of the air-coupled transducer to the projection of the air-coupled transducer on the $x_{2}y_{2}z_{2}$ coordinate system, $k_{p2}=\omega /c_{p2}$ is the wavenumber in the air, $S^{\prime }$ is the projected area of the surface of the air-coupled transducer on the surface of the specimen, and $S_{r}$ is the area of the air-coupled transducer. Since the distance between the air-coupled transducer and the specimen is fixed, only the diffraction effect of the Rayleigh wave propagating in the specimen needs to be considered, which can be expressed by multi-Gaussian beam, as [22],
(2)$$\begin{array}{l} u^{D}\left(x\right)=\frac{A_{m}}{1+iB_{m}z_{2}/D_{R}}\frac{1}{\sqrt{M_{1}}\sqrt{M_{2}x_{2}+k_{r}}}\exp\left(-\frac{ik_{r}y_{2}^{2}}{2}\left(\frac{M_{2}}{M_{2}x_{2}+k_{r}}\right)\right)\\ M_{1}=\frac{k_{p1}\cos^{2}\theta_{1}}{z_{1}-iD_{R}/B_{m}},\;M_{2}=\frac{k_{p1}}{z_{1}-iD_{R}/B_{m}} \end{array}$$
where $A_{m}$ and $B_{m}$ are 25 groups of Gaussian coefficients [23], $D_{R}=k_{r}a^{2}/2$ is the Rayleigh distance, a is the radius of the angle beam wedge transducer. As shown in Equations (1) and (2), when the wave propagation distances in the wedge and air are fixed and diffraction effects are considered, only the attenuation of the Rayleigh wave affects the distribution characteristics at different propagation distances. Therefore, the attenuation coefficient $\alpha_{r}$ can be extracted using the least squares curve fitting method as,
(3)$$\min_{u_{0,1},\alpha_{r}}\|u_{1}\left(\left\{u_{0,1},\alpha_{r},\right\},x\right)-u_{1}^{MEAS}{\|}_{2}^{2}=\min_{u_{0,1},\alpha_{r}}{\sum_{i}\left[u_{1}\left(\left\{u_{0,1},\alpha_{r}\right\},x_{i}\right)-u_{1,i}^{MEAS}\right]}^{2}$$
where $u_{1}$ represents the wave displacement function. The arguments to $u_{1}$ include the values of the fitting parameters $\left\{u_{0,1},\alpha_{r}\right\}$ and the propagation distance x. $u_{0,1}$ represents the scaling parameter of displacement and voltage. While, $u_{1}^{MEAS}$ represents the measured voltage amplitude after diffraction correction at their respective frequencies. It needs to be noted that there is a certain linear relationship between the voltage value and the displacement amplitude, which will not affect the fitting of the parameters [24]. The subscript i on the right side of the equation indicates discrete data, which is related to the data obtained in the experiments.

The fitting process is shown in Figure 2, and the detailed process is as follows: (a) A Hanning window (black) is used to filter the received wave signals, and the windowed time-domain signal is transferred to a frequency-domain signal using a fast Fourier transformation (FFT). (b) The amplitude at the driving frequency is obtained as the wave displacement, $u_{1}$. (c) The measured voltage amplitude of the diffraction correction using Equation (2) at different propagation distances are obtained. (d) Equation (1) is used for the fitting process to extract $u_{0,1}$ and $\alpha_{r}$.

## 3. Experiments

### 3.1. Specimen Preparation

The chemical composition and mechanical properties of 316L stainless steel at room temperature of the investigated 316L stainless steel is provided by a commercial company and shown in Table 1 and Table 2 for reference. All specimens with dimensions 50 × 20 × 230 mm^3^ were prepared by cutting from a 316L stainless steel plate and were heat treated as shown in Figure 3. These specimens were solution heat-treated at 540 °C for 240 min, and subsequently quenched in water for 60 min. The quenched specimens were then heat-treated at 220 °C for various periods of time: 0, 30, 60, 120, 300, and 600 min. The surface with a size of 50 × 230 mm^2^ was polished with different grades of sandpaper and the opposite surface remained unchanged. After the ultrasonic testing was completed, the suitable size samples were cut from the near surface of the specimens. The samples were polished and etched for 20 min using an etchant of 20% HF + 10% HNO_3_ + 70% H_2_O [2].

### 3.2. Ultrasonic Experimental Process

The experimental system for measuring the ultrasonic attenuation coefficient of the Rayleigh waves is shown in Figure 4. A 10-cycle tone-burst wave signal with a center frequency of 2MHz and an amplitude of 400 mV voltage was generated by a function generator (33250A, Agilent Technologies, Inc., Santa Clara, CA). This signal was amplified by 50 dB using an Amplifier (2100L, Electronics and Innovation, Ltd., Rochester, NY, USA) and applied to the angle beam wedge transducer (V404, OLYMPUS., Waltham, MA, USA). The Rayleigh wave was generated and propagated a certain distance along the surface of the specimen and finally received by an air-coupled transducer (NCT4-D13, Ultran Group., State College, PA), then converted into an electrical signal. This electrical signal was amplified by 55 dB with a pre-amplifier (DPR300, Ultrasonics, JSR., Pittsford, NY, USA) and digitized by a Waverunner oscilloscope (MDO3024, Tektronix, Inc., Wilsonville, OR, USA). The distance between the center of the air coupled transducer surface and the projection point of the specimen surface was fixed at 6 mm. The tilt angle of the air-coupled transducer, which is determined by the theoretical Rayleigh wave velocity in the steel and the longitudinal wave velocity in air was adjusted to 6° according to the scale on the fixture. It should be noted that this angle is slightly adjusted around 6° to compensate for variations in the Rayleigh wave velocity for each specimen. The Rayleigh waves were measured in the propagation range of 20~120 mm with a gap of 5 mm. Every experiment was repeated 5 times.

## 4. Results and Discussions

### 4.1. Microstructure Properties of the 316L Steel Specimens

The metallographic images of the samples were obtained using a Leica DM4000M microscope system, and are shown in Figure 5. It is observed that the grain size is about 25 μm at the initial state: New grains with bigger sizes will be generated after heat treatment, or the grains will grow as the heat treatment time increases. Therefore, the grain size of the specimens is bigger than that in the specimen at the initial state, as shown in Figure 5b–f. The average grain size of the samples in Table 3 were measured according to the standard of ASTM E112 [25].

### 4.2. Measurement Precision Using an Air-Coupled Receiver

The time domain signal, received by the air-coupled transducer, is shown in the Figure 6. It can be seen from the figure that the signal has a very high signal-to-noise ratio (44 dB) after being amplified by the preamplifier and averaging by the oscilloscope. Based on this, it can be considered that the influence of noise on the measurement uncertainty can be ignored.

The surface state of the material and the coupling state between receiver and specimen affect the propagation and reception of Rayleigh waves because the energy of Rayleigh waves is mainly concentrated on the surface of the material. To quantitatively study these effects, the measurements were performed using an air-coupled receiver and a contact receiver on the polished and unpolished surfaces of the steel specimens. In the measurement process, the angle bean wedge transducer was used as the transmitter and light lubrication oil was used as the coupling. The measured voltage amplitude results in unpolished and polished specimen using angle beam wedge transducer as receiver are shown in Figure 7a,b respectively. The measured voltage amplitude results in unpolished and polished specimen using air-coupled transducer as receiver are shown in Figure 7c,d respectively.

As the experiments were conducted at the unpolished surface of the specimen, the results of the voltage amplitude (Figure 7a) using an angle beam wedge receiver fluctuated greatly and the error bar was large, while the results obtained using an air-coupled receiver are much better (Figure 7c). The errors of measurements at different surface conditions, and using different receivers, are shown in Table 4. The maximum error is the maximum value of the relative error among all points, and the minimum error is the minimum value of the relative error among all points. The average variance is the average of the variance of each measurement distance. In particular, it can be seen from Table 4 that the errors of the repeated measurements, using an air-coupled receiver, are much smaller than those using an angle beam wedge receiver. This indicates that the reception of the air-coupled transducer is more stable. Moreover, the same conclusion can be drawn for the polished surface.

By comparing Figure 7a with Figure 7b, and Figure 7c with Figure 7d, it can be seen that the measured amplitudes at the polished surface have narrow error bars. Moreover, as seen in Table 4, regarding the angle beam wedge receiver, the maximum error has declined by 6.99% and the average variance has declined by 0.81 mV^2^ when the measurements are conducted at the polished surfaces instead of the unpolished ones. As the air-coupled receiver is used, the maximum error declined by 1.22% and the average variance declined by 0.13 mV^2^. Furthermore, when using an angle beam wedge receiver, the errors vary greatly at both the polished and unpolished surfaces. These results indicate that measurement using an air-coupled receiver is less affected by the surface conditions.

The measured and predicted results for the extracted attenuation coefficient are shown in Figure 8. It can be seen that when an angle beam wedge receiver is used the extracted attenuation coefficient at the polished surface is less than that at the unpolished surface (from 1.88 Np/m to 1.47 Np/m). However, the measured attenuation coefficients are almost the same (from 1.5 Np/m to 1.45 Np/m) when an air-coupled receiver is used. In addition, the value of the correlation coefficients between the theoretical and the experimental, measured using an air-coupled receiver, is larger than that using an angle wedge receiver. These results show that measurements using an air-coupled receiver have a better precision.

### 4.3. Measurement of the Attenuation Coefficient for Heat Treatment Stainless Steel

The attenuation coefficients of the Rayleigh waves for the heat treated 316L stainless steel are measured using the method described at Section 2. In the experiments, the air-coupled transducer works as the receiver, and the measurements are conducted at the unpolished surfaces. Figure 9 shows the nonlinear curve fitting results for one experiment. The normalized variation curve of the Rayleigh wave attenuation coefficient, hardness and grain size with heat treatment time is shown in Figure 10. The hardness can reflect the macroscopic properties of the material to a certain extent, and the normalized result is also shown in the Figure 10. The normalization process is based on the values of the three indicators at 0 min heat treatment time.

As seen from Figure 10, as the heat treatment time increases, both the attenuation coefficient and the grain size have an upward trend, but the hardness decreases. Moreover, the tendency for rising or decreasing is quite slow in the early stage of heat treatment but becomes significant in the later stage. It is found that scattering and absorption are the main factors to affect the wave attenuation. Absorption is caused by microstructural defects such as dislocations and point defects, which have little impact on the attenuation and can usually be ignored. By increasing the heat treatment time, the formation and growth of the precipitates happen, grain sizes will increase, and wave scattering is enhanced, which leads to an increase in the attenuation [26]. The increase in the grain size will reduce the mechanical properties of the 316L steel. One of the performance degradation behaviors is that the hardness decreases [27]. The experimental results show that the Rayleigh wave attenuation coefficient can effectively evaluate the change of grain size and the mechanical property of the steel.

In order to validate the effectiveness of the material characterization using the attenuation coefficient of the Rayleigh waves, the attenuation coefficient of the longitudinal waves was also measured to characterize these heat-treated 316L stainless steels. The attenuation was measured using a pulse-echo experiment [28], and transducers which have central frequencies of 2 MHz, 3.5 MHz, and 7 MHz were used. The attenuation coefficients of the longitudinal wave for each specimen were measured five times and the results are shown in Table 5. The variation curve of the longitudinal wave attenuation coefficient, at different frequencies with heat treatment time, is shown in Figure 11. It can be seen that the results at 2 MHz are chaotic and irregular, which indicates that the longitudinal wave’s attenuation coefficient cannot be accurately measured at low frequencies for characterizing the grain size of heat-treated stainless steel. The results measured at the frequency of 3.5 MHz and 7 MHz have an increasing tendency with aging time. This is because high frequency ultrasonic waves have relatively shorter wavelengths and show more sensitivity to interact with fine precipitates [26]. However, even at a low frequency (2 MHz), the variation of the Rayleigh wave attenuation coefficient is sufficient to distinguish the changes of the material properties of the heat-treated stainless steel. This is due to Rayleigh wave has shorter wavelength at the similar frequency than longitudinal wave, meaning that it is more sensitive than the longitudinal wave.

## 5. Conclusions

In this work, a method is presented to measure the Rayleigh wave attenuation using an air-coupled receiver, and show its feasibility and superiority for characterizing grain size of materials. The different grain size of 316L stainless steel was obtained by heat treatment. It was found that when the grain size increased due to heat treatment time, the attenuation coefficient of the Rayleigh waves increased and the hardness decreased accordingly. At similar driving frequencies, Rayleigh wave attenuation varies significantly, while there is no obvious change of the longitudinal wave attenuation. This indicate that the Rayleigh wave attenuation coefficient can effectively indicate the change of grain size. Compared with the measurements using an angle beam wedge receiver, the measurements are efficient, more stable and less influenced by the surface state when an air-coupled receiver is used, which shows that the proposed method is more practical. In order to realize the quantitively characteristic of grain size with the Rayleigh wave attenuation, corresponding scattering theory, related to Rayleigh waves, and the influence of anisotropy in steel on the Rayleigh attenuation coefficient measurement will be explored in the future.

## Figures and Tables

**Figure 1 materials-14-01901-f001:**
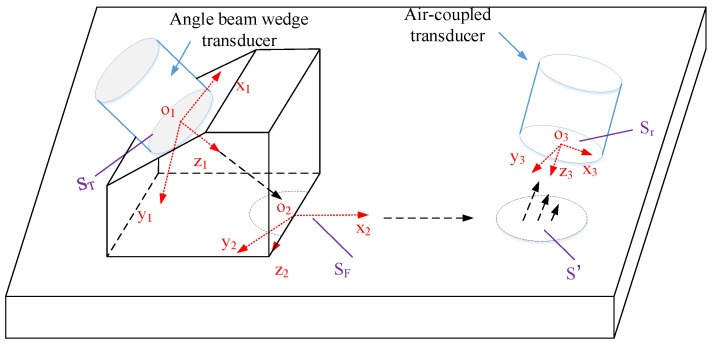
Schematic diagram of Rayleigh wave generation, propagation, and reception.

**Figure 2 materials-14-01901-f002:**
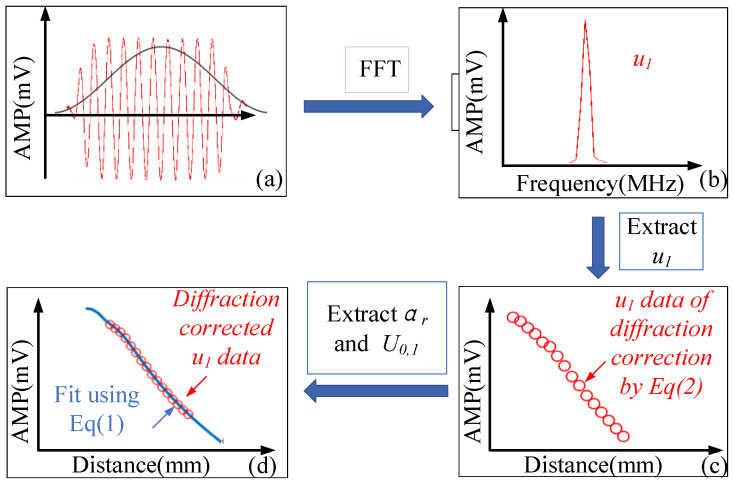
Attenuation coefficient fitting process. (**a**) A Hanning window (black) to filter the received wave signal. (**b**) The frequency-domain signal after fast Fourier transformation (FFT). (**c**) The measured voltage amplitude of the diffraction correction at different propagation distances. (**d**) The fitting process to extract $u_{0,1}$ and $\alpha_{r}$.

**Figure 3 materials-14-01901-f003:**
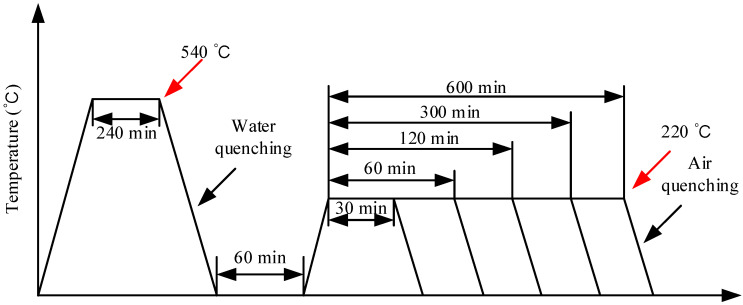
Heat treatment process diagram.

**Figure 4 materials-14-01901-f004:**
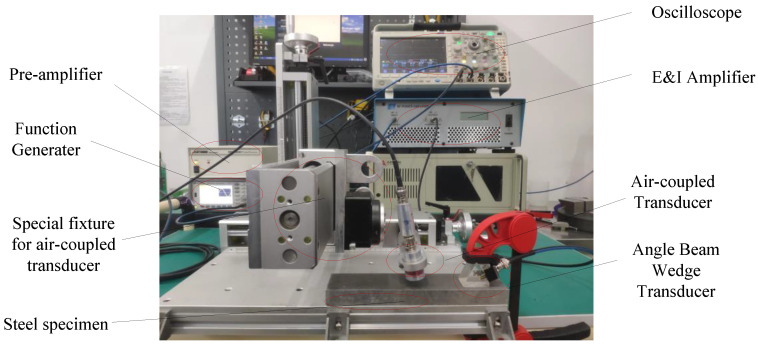
Schematic diagram of Rayleigh wave attenuation measurement setup.

**Figure 5 materials-14-01901-f005:**
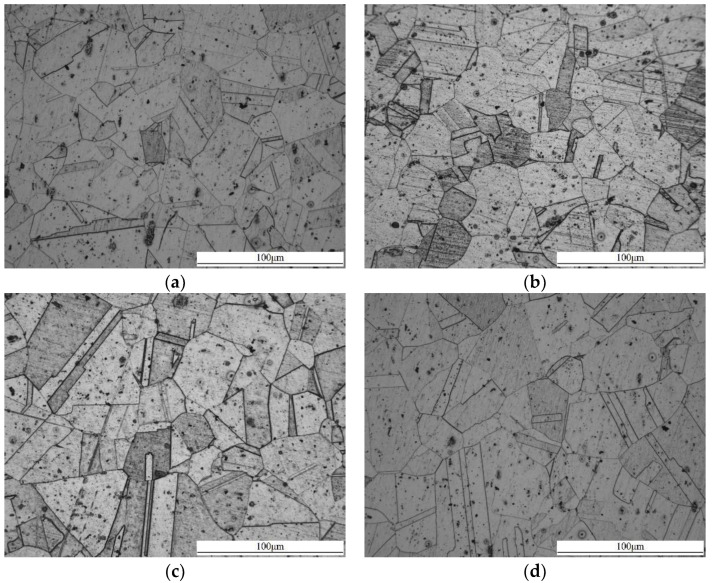
Microstructural evolution of 316L stainless steel at six different stages: (**a**) Heat treatment time 0 min. (**b**) Heat treatment time 30 min. (**c**) Heat treatment time 60 min. (**d**) Heat treatment time 120 min. (**e**) Heat treatment time 300 min. (**f**) Heat treatment time 600 min.

**Figure 6 materials-14-01901-f006:**
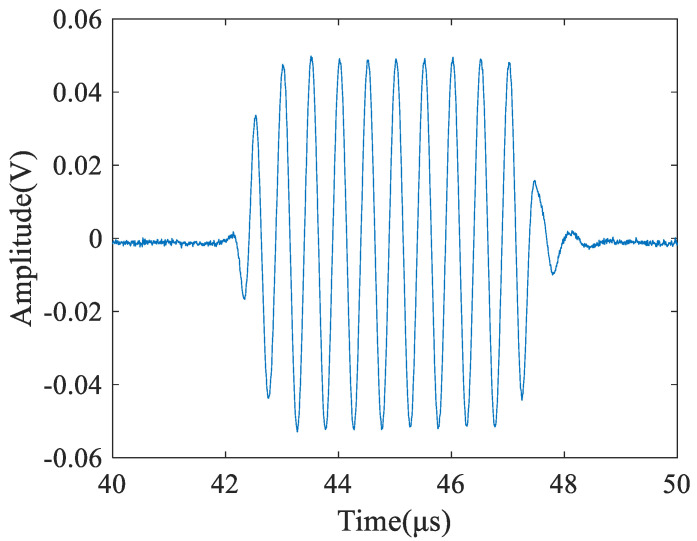
A time domain signal received by the air-coupled transducer.

**Figure 7 materials-14-01901-f007:**
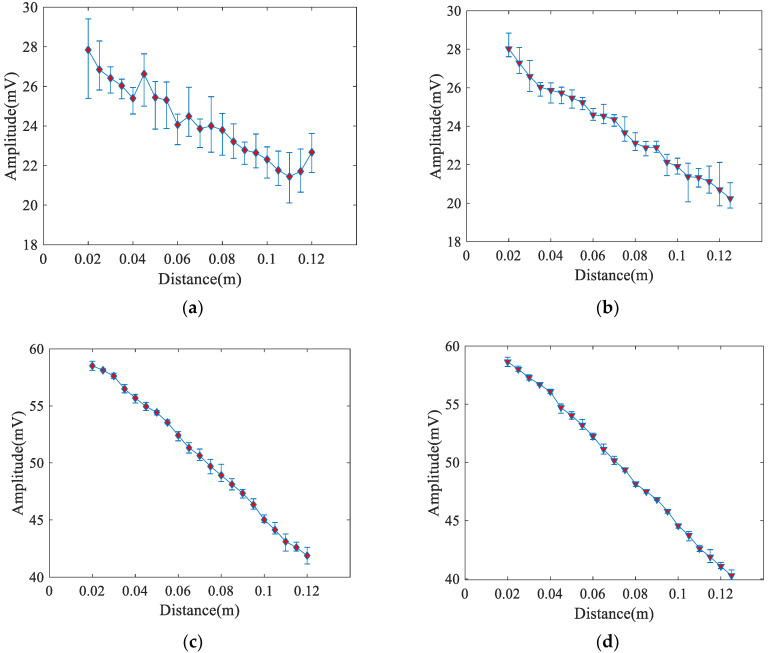
Average voltage amplitude. (**a**) Angle beam wedge receiver and unpolished surface. (**b**) Angle beam wedge receiver and polished surface. (**c**) Air-coupled receiver and unpolished surface. (**d**) Air-coupled receiver and polished surface.

**Figure 8 materials-14-01901-f008:**
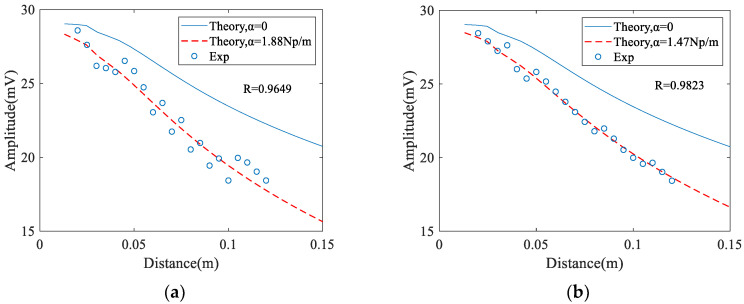
Experimental measurement results and least squares fitting curve: (**a**) Angle beam wedge receiver and unpolished surface. (**b**) Angle beam wedge receiver and polished surface. (**c**) Air-coupled receiver and unpolished surface, (**d**) Air-coupled receiver and polished surface.

**Figure 9 materials-14-01901-f009:**
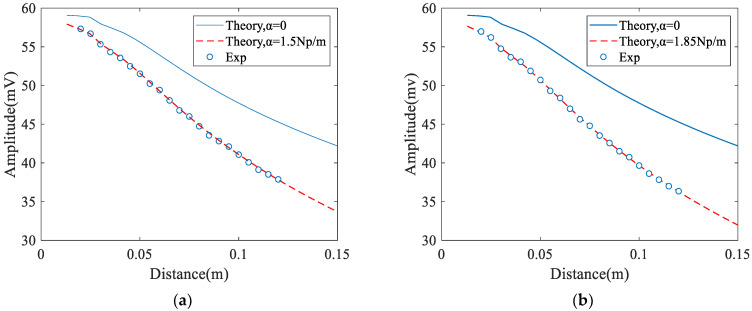
Experimental and theoretical prediction results of Rayleigh wave attenuation coefficients with different heat treatment times. (**a**) 0 min. (**b**) 30 min. (**c**) 60 min. (**d**) 120 min. (**e**) 300 min. (**f**) 600 min.

**Figure 10 materials-14-01901-f010:**
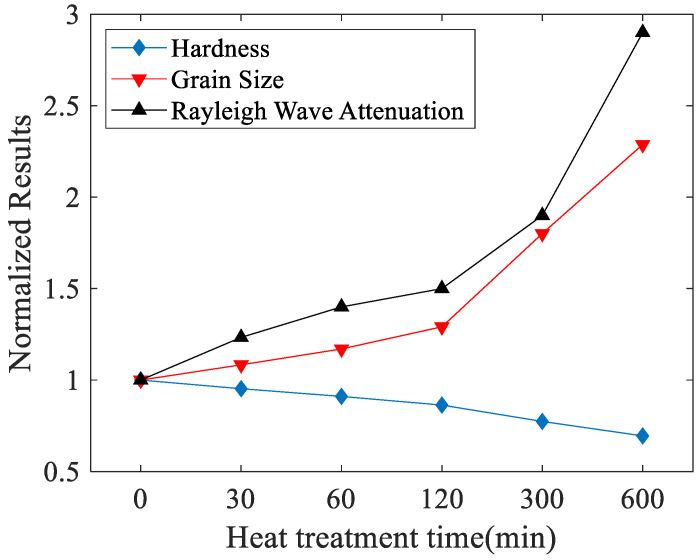
Variation curve of Rayleigh wave attenuation coefficient (Np/m), hardness (HV), and grain size (μm) with heat treatment time.

**Figure 11 materials-14-01901-f011:**
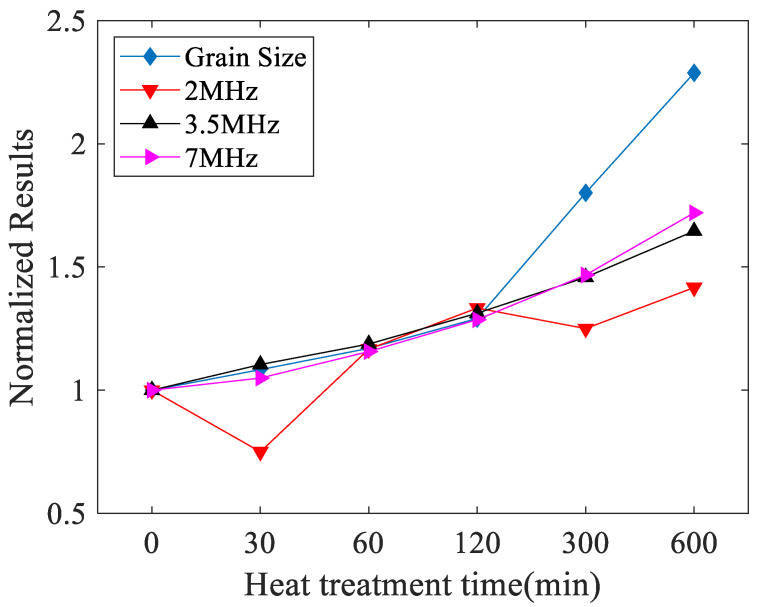
Variation curve of longitudinal wave attenuation coefficient (Np/m) and grain size (μm) with heat treatment time.

**Table 1 materials-14-01901-t001:** Chemical composition of 316L stainless steel (wt.%).

C	Cr	Ni	Mo	Mn	P	S	Si
≤0.03	16–18	10–14	2–3	≤2.00	≤0.045	≤0.03	≤1.00

**Table 2 materials-14-01901-t002:** Mechanical properties of 316L stainless steel at room temperature.

Yield Strength (MPa)	Tensile Strength (MPa)	Elastic Modulus (MPa)	Poisson’s Ratio	Hardness (HV)
269	603	206	0.3	190

**Table 3 materials-14-01901-t003:** Average grain size of samples with different heat treatment time.

Sample	Average Grain Size (μm)
(a)	24.11 ± 1.59
(b)	26.12 ± 0.73
(c)	28.21 ± 1.83
(d)	31.13 ± 0.99
(e)	43.42 ± 2.31
(f)	55.16 ± 4.50

**Table 4 materials-14-01901-t004:** Errors of measurement results shown in Figure 7.

Number	Maximum Error (%)	Minimum Error (%)	Average Variance (mV^2^)
(a)	13.87	2.23	1.21
(b)	6.88	0.92	0.40
(c)	2.93	0.18	0.22
(d)	1.71	0.13	0.09

**Table 5 materials-14-01901-t005:** Longitudinal wave attenuation coefficient measurement results.

Sample	2 MHz (Np/m)	3.5 MHz (Np/m)	7 MHz (Np/m)
(a)	1.2 ± 0.4	4.8 ± 0.6	32.5 ± 2.5
(b)	0.9 ± 0.6	5.3 ± 0.8	34.1 ± 3.2
(c)	1.4 ± 0.3	5.7 ± 0.8	37.6 ± 4.2
(d)	1.6 ± 0.4	6.3 ± 1.2	41.8 ± 4.9
(e)	1.5 ± 0.5	7.0 ± 1.2	47.7 ± 4.6
(f)	1.7 ± 0.7	7.9 ± 0.9	55.9 ± 5.4

## Data Availability

Not applicable.

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
