# Peer review of "Characterization of Grain Size in 316L Stainless Steel Using the Attenuation of Rayleigh Wave Measured by Air-Coupled Transducer"

_materials, 2021, doi:10.3390/ma14081901_

Round 1

Reviewer 1 Report

Remarks to the authors (take a look at the file)

Reviewer 2 Report

The article is globally of good quality and its structure is easy to read. The proposed method seems efficient and interesting. However, there are a lot of english errors which should be corrected to improve the readability of the article (I did not pinpoint them, but it seemed that there were more in the introduction than later in the text). I also have a few remarks which I think could improve the quality of the paper:

  • eq (3): the minimization should be over u_0,1 and \alpha_r, not x
  • p.6, l162: a ref to ASTM E112 would be nice
  • p.7, l172 and figure 6: There is something weird here: are you doing measurements with wedges for both emission and reception as well as measurements with an air-coupled transducer as receiver? If so, it needs to be specified in the experimental protocol. If not, I do not understand figure 6 and something needs to be clarified. On which plate are conducted these measurements?
  • p.7 l183: table2
  • Table2: how is this error computed, do you use a reference to obtain it?
  • p.10 l239: table3
  • p.10 l245: for a more convincing conclusion, you could compare the evolution of the longitudinal wave attenuation coefficient wrt to the grain size to that of Rayleigh waves in figure 9

Reviewer 3 Report

Remarks are in the file.

Reviewer 4 Report

The paper reports about “Measurement of Rayleigh wave attenuation using an air-coupled transducer to characterize grain size in 316L stainless steel”. The topic of the reviewed article is a very interesting for Researchers dealing with materials science. However, the manuscript requires a minor revision prior to publication.

1.The following suggestions and comments have to be addressed before publication of the paper:

2.In Figure 4, the applied indicators showing the elements of Rayleigh wave are poorly visible, please correct this.

3.The text fragment such as “After the ultrasonic testing was completed, the suitable size samples were cut from the near surface of the specimens. The samples were ground, polished, and etched for 20 min using an etchant of 20% HF + 10% HNO3 + 70% H2O [2]”, should be included by part of the Experiments / Specimen preparation or Ultrasonic experimental process.

4.Please change a location of the Figure 7, after paragraph starting “The measured and predicted results for the extracted attenuation coefficient are 203 shown in Fig. 7”.

5.In the Subchapter 3.1, please add the chemical composition and properties (mechanical and thermophysical) of the 316L steel.

6.In the Chapter 4, please indicate effect of the properties of the 316L steel for the results analysis.
